# Weight Change during the Early Phase of Convalescent Rehabilitation after Stroke as a Predictor of Functional Recovery: A Retrospective Cohort Study

**DOI:** 10.3390/nu14020264

**Published:** 2022-01-09

**Authors:** Hiroshi Kishimoto, Yuka Nemoto, Takayuki Maezawa, Kazushi Takahashi, Kazunori Koseki, Kiyoshige Ishibashi, Hanako Tanamachi, Naoki Kobayashi, Yutaka Kohno

**Affiliations:** 1Department of Physical Medicine and Rehabilitation, Ibaraki Prefectural University of Health Sciences Hospital, Ibaraki 300-0331, Japan; 2Department of Nutritional Management, Ibaraki Prefectural University of Health Sciences Hospital, Ibaraki 300-0331, Japan; negisan0523@gmail.com; 3Department of Physical Therapy, Ibaraki Prefectural University of Health Sciences Hospital, Ibaraki 300-0331, Japan; maesawat@ami.ipu.ac.jp (T.M.); takahashik@ami.ipu.ac.jp (K.T.); koseki@ami.ipu.ac.jp (K.K.); ishibashik@ami.ipu.ac.jp (K.I.); takanoh@ami.ipu.ac.jp (H.T.); kobayashin@ami.ipu.ac.jp (N.K.); 4Department of Neurology, Ibaraki Prefectural University of Health Sciences Hospital, Ibaraki 300-0331, Japan; kohno@ipu.ac.jp

**Keywords:** convalescent rehabilitation, stroke, body weight, functional recovery, nutritional management

## Abstract

It has been reported that weight gain at discharge compared with admission is associated with improved activities of daily living in convalescent rehabilitation (CR) patients with low body mass index. Here, we investigated whether weight maintenance or gain during the early phase of CR after stroke correlates with a better functional recovery in patients with a wide range of BMI values. We conducted this retrospective cohort study in a CR ward of our hospital and included adult stroke patients admitted to the ward from January 2014 to December 2018. After ~1 month of hospitalization, the patients were classified into weight loss and weight maintenance or gain (WMG) groups based on the Global Leadership Initiative on Malnutrition criteria for weight. We adopted the motor functional independence measure (FIM) gain as the primary outcome. The motor FIM gain tended to be greater in the WMG group but without statistical significance. However, multiple regression analysis showed that WMG was significantly and positively associated with motor FIM gain. In conclusion, weight maintenance or gain in patients during the early phase of CR after stroke may be considered as a predictor of their functional recovery, and nutritional management to prevent weight loss immediately after the start of rehabilitation would contribute to this.

## 1. Introduction

The aging of the society is a global challenge today, and frailty [1] has become a critical issue. While there have been many molecular biological [2] and biochemical [3,4] studies on frailty, various epidemiological studies have also been conducted, and it is known that people with frailty have a higher risk of developing strokes [5], as well as a higher risk of falls and hip fractures [6]. In Japan, the national health insurance system established a convalescent rehabilitation (CR) ward in the year 2000, which has played an important role in the post-acute care of patients with stroke, brain or spinal cord injury, hip fracture, and hospital-associated deconditioning [7]. It has been reported that the prevalence of malnutrition, malnutrition risk status, and sarcopenia is high among patients admitted to the ward [8,9]. In addition, having malnutrition or sarcopenia has been associated with poor recovery of physical function in CR [9,10]. On the other hand, improvement of nutritional status among malnourished elderly patients with stroke during CR has been linked to improved activities of daily living (ADLs) [11,12]. Furthermore, we have previously reported that a group of patients whose nutritional status was maintained at good or even slightly improved from poor during CR had better functional recovery than a group of patients whose nutritional status remained poor or worsened even if their status was good at admission [13].

Among these studies, MNA -SF [14] in Ref. 11 GNRI [15] in Ref. 12, and CONUT [16] in Ref. 13 have been used as diagnostic tools for malnutrition or monitoring indicators of nutritional status, and recently, “GLIM(Global Leadership Initiative on Malnutrition) criteria for the diagnosis of malnutrition” [17] has been proposed as a consensus report from several clinical nutrition societies around the world. The GLIM criteria state that screening for nutritional status should be conducted using validated tools such as the MNA-SF [14], NRS-2002 [18], MUST [19], and SGA [20] and that priority should be given to repeated weight measurements over time to identify trajectories of weight loss, maintenance, and improvement. The importance of recognizing the pace of weight loss in the early stages of illness or injury has been emphasized in GLIM criteria [17]. One study that focused on weight change in CR was conducted by Kokura et al. [21]. They reported that in CR patients with a low body mass index (BMI) at admission, weight gain over the entire hospital stay up to the time of discharge was associated with improved ADL [21].

As in the case of frailty, molecular biological analysis has been conducted on stroke patients [22], but many epidemiological studies have also been conducted [23,24,25]. The range of overweight not reaching obesity is also considered to be a risk factor for stroke [24], and it has been shown that there are not a few stroke rehabilitation patients with high BMI [25]. However, the relationship between weight change in CR and improvement of ADL at the time of CR discharge in patients with a wide range of BMI has not been clarified so far. In addition, we have not found any studies that have assessed weight change in the early stages of CR and examined its relationship with functional recovery.

This study aims to address the clinical question of whether weight maintenance gain or loss in the early stages of CR in patients with stroke has a positive or negative impact, respectively, on functional recovery at discharge.

## 2. Materials and Methods

### 2.1. Participants and Setting

We conducted a retrospective cohort study in a 47-bed CR ward of our hospital in Japan. Adult stroke patients admitted to the ward from January 2014 to December 2018 were included in the study. In this ward, the daily program consisted of physical therapy (PT), occupational therapy (OT), and speech and hearing therapy (ST), with a maximum of 180 min per day (according to the Japanese health insurance system). On average, about 150 min were provided per day, including 40–120 min of PT, 40–120 min of OT, and 0–60 min of ST. The training content is the same as in previous literatures [26,27]. Patients with a hospital stay of fewer than 30 days after cohorting were excluded due to a short observation period. We excluded patients with a BMI of 30 or higher on admission because they are treated as obese under the Japanese health insurance system and are offered a diet aimed at weight loss. Patients with missing data were also excluded.

### 2.2. Cohorting

Based on the GLIM criteria for weight loss, “unintentional weight loss of >5% within the past 6 months” [17], the weight at admission was subtracted from the weight on day N, ~1 month after admission, and then multiplied by 180/N to convert to a change per 180 days. Patients whose converted decrease was >5% of their original weight were classified as the weight loss (WL) group and the remaining patients formed the weight maintenance or gain (WMG) group.

### 2.3. Data Collection

Baseline patients’ characteristics and parameters such as age, gender, BMI, stroke type, number of days between the onset and admission to our rehabilitation ward, serum albumin level, serum creatinine level, and motor/cognitive functional independence measure (FIM) scores at admission [28] were obtained from the medical records retrospectively. The presence of dysphagia was defined by tube feeding or provision of the texture-modified meal to support the swallowing function. The severity of complications was assessed using the Charlson comorbidity index (CCI) [29]. Energy intake (EI; kcal) was determined from the ratio of the patient’s food intake to the food supply recorded visually by the nurse or dietitian and averaged over the three days prior to cohorting. The ratio of EI and basal energy expenditure (BEE), calculated from the Harris–Benedict equation [30], was also recorded for each patient. The number of days spent in the hospital and the rehabilitation minutes per day during the hospital stay were also recorded.

### 2.4. Outcome

We adopted the motor FIM [28] gain as the primary outcome of the study. The FIM consists of 13 motor items and five cognitive items. Each item is rated on a scale of 1–7, with motor items scored from 13 to 91 and cognitive items scored from 5 to 35. The motor FIM gain was calculated as the difference in motor FIM score between the discharge and admission.

### 2.5. Sample Size Calculation

Statistical software G*power 3 [31] was used for sample size calculation.

Multiple regression analysis using FIM efficiency as the outcome in our previous study [13] yielded an effect size of f2 = 0.159. Using the effect size of 0.159, significance level of α = 0.05, power of test 1 − β = 0.9, assuming the number of planned explanatory variables to be 13, the required sample size was calculated to be 153. Since the number of patients admitted to our CR ward has ranged from 60 to 90 per year, we decided to obtain statistics for 5 years by considering the exclusion criteria and missing data.

### 2.6. Statistical Analysis

Mann–Whitney *U* test, *t* test, chi-square test, and Fisher’s exact probability test were used to compare the two groups, depending on the type of variable. The Shapiro–Wilk test was performed to assess normality. To examine the association between WMG/WL and the outcome, we applied multiple regression analysis. To avoid the effects of confounding variables, additional explanatory variables included age, sex, type of stroke, number of days since onset, BMI, serum albumin level, serum creatinine level, motor FIM at admission, CCI, presence of dysphagia, duration of hospital stay, and rehabilitation time per day. Since EI/BEE was expected to be strongly related to WMG, we performed an additional multiple regression analysis (Model 2) using EI/BEE instead of WMG. Multicollinearity was assessed by the variance inflation factor (VIF), and if the VIF value was less than 10, it was considered that no multicollinearity was observed. The significance level was set at a *p*-value < 0.05 and all the analyses were performed using statistical software SPSS Statistics, version 26 (IBM Japan, Tokyo Japan).

## 3. Results

Of the 393 patients admitted to the CR ward during the 5-year period, 100 were excluded from the study, resulting in 293 patients for analysis, divided into the WMG group (*n* = 176) and the WL group (*n* = 117) (Figure 1). Table 1 shows the patient characteristics. There were significant differences between the two groups in age, number of days from onset to admission to the convalescent rehabilitation ward, BMI, serum albumin level at admission, EI, and EI/BEE. The motor FIM gain as the outcome of the study was greater in the WMG group, but it did not reach statistical significance.

The results of multiple regression analysis with motor FIM gain as the dependent variable and WMG as an explanatory variable showed that WMG was significantly and positively associated with motor FIM gain (standardized coefficient = 0.105, *p* = 0.043, adjusted R-square = 0.320). In addition, age, days from onset to admission to the convalescent rehabilitation ward, motor FIM at admission, CCI, and dysphagia showed significant negative associations, whereas BMI and hospital stay showed significant positive associations (Table 2).

Multiple regression analysis (model 2, Table 3), which included motor FIM gain as the outcome and EI/BEE instead of WMG as an explanatory variable, showed that EI/BEE was significantly and positively associated with motor FIM gain (standardized coefficient = 0.169, *p* = 0.005, adjusted R-square = 0.329). In both models, no variable had a VIF > 10.

## 4. Discussion

In comparing the two groups, the WMG group had a greater motor FIM gain than the WL group, but the difference was not statistically significant. However, after adjusting for confounding factors, such as age, sex, type of stroke, number of days since onset, BMI, serum albumin level, serum creatinine level, motor FIM at admission, Charlson comorbidity index, presence of dysphagia, duration of hospital stay, and rehabilitation time per day, in a multiple regression analysis, WMG showed a significant positive association with motor FIM gain.

Instead of weight gain during the hospitalization period of convalescent patients in the previous study [21], our study showed that WMG during the early stage of convalescent rehabilitation after stroke was associated with better functional recovery. It is also noteworthy that the study included not only underweight patients but also patients with a wide range of body weight, excluding only obese patients who needed weight loss guidance. Although it is unknown whether functional outcomes in adults undergoing inpatient stroke rehabilitation are affected by obesity [25], in this study, BMI was significantly positively associated with motor FIM gain in the group of patients with a BMI < 30.

Multiple regression analysis (model 2), where EI/BEE was used as an explanatory variable instead of WMG, revealed that EI/BEE was also significantly positively associated with motor FIM gain. These results are supported by Nii et al. who reported that nutritional intake in the early stage of admission to a rehabilitation ward is associated with FIM efficiency [12]. Shimazu et al. reported that frequent individualized nutritional support was associated with improved nutritional status, physical function, and dysphagia after stroke, which emphasizes the need for intensive multidisciplinary nutritional support [32]. These findings and the results of our study suggest that in stroke convalescent rehabilitation, frequent nutritional support to maintain or increase weight should be provided by the team in charge of each patient, ensuring adequate EI by constantly considering the increased energy expenditure to rehabilitation. In such support, the median EI/BEE of 1.30 in the WMG group could be considered as one of the guidelines for energy supply in stroke rehabilitation.

Besides being retrospective and single-center study, our study is also limited by the fact that motor FIM gain was used as the outcome, so functional recovery was only captured from aspects that can be assessed by motor FIM, and that muscle mass change may be more strongly associated with motor FIM gain compared with weight change. Although body weight measurements are more readily available compared with muscle mass measurements, a recent study [33] has shown that muscle mass gain could be positively associated with functional recovery in patients with sarcopenia after stroke. Future studies should focus on muscle mass change as well.

## 5. Conclusions

In conclusion, weight maintenance or gain in patients during the early phase of hospitalized convalescent rehabilitation after stroke was a predictor of functional recovery in patients with a BMI of <30. Therefore, nutritional management to prevent weight loss immediately after the start of convalescent rehabilitation is advised to ensure better functional recovery in these patients.

## Figures and Tables

**Figure 1 nutrients-14-00264-f001:**
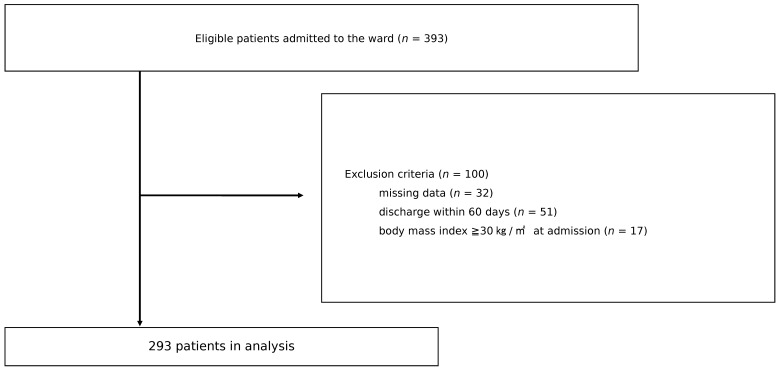
Study flowchart.

**Table 1 nutrients-14-00264-t001:** Patients’ characteristics.

	Total (*N* = 293)	WMG (*N* = 176)	WL (*N* = 117)	*p* Value	
Age, years, median [IQR]	69 [60–78]	70 [61–79]	67 [56–76.5]	0.041	*
Sex: Male, *n* (%)	178 (60.8)	99 (55.6)	79 (44.4)	0.067	†
Female, *n* (%)	115 (39.2)	77 (67.0)	38 (33.0)		
Stroke type					
Cerebral infarction, *n* (%)	151 (51.5)	85 (56.3)	66 (43.7)	0.333	‡
Intracerebral hemorrhage, *n* (%)	119 (40.6)	75 (63.0)	44 (37.0)		
Subarachnoid hemorrhage, *n* (%)	23 (7.8)	16 (69.6)	7 (30.4)		
Days from onset to admission, d, median [IQR]	36 [27.5–50]	38 [30–51.75]	35 [24–42.5]	0.007	*
Days from admission to cohorting, d, median [IQR]	28 [21–35]	28 [21.75–35]	28 [21–34]	0.482	*
Serum albumin level on admission, mg/dL, median [IQR]	3.9 [3.5–4.2]	3.85 [3.4–4.2]	4.1 [3.7–4.3]	0.001	*
Serum creatinine level on admission, mg/dL, median [IQR]	0.7 [0.6–0.9]	0.7 [0.6–0.9]	0.7 [0.6–0.9]	0.321	*
CCI, median, [IQR]	2 [2–3]	2 [2–3]	2 [2–3]	0.969	*
BMI at admission, kg/m^2^, mean (SD)	22.1 ± 3.1	21.4 ± 3.0	23.0 ± 2.9	<0.001	§
Low BMI at admission (GLIM criteria for Asians)					
Yes, *n*, (%)	50 (17.1)	40 (22.7)	10 (8.5)	0.001	†
No, *n*, (%)	243 (82.9)	136 (77.3)	107 (91.5)		
BMI at discharge, kg/m^2^, mean (SD)	21.7 ± 2.7	21.6 ± 2.8	21.8 ± 2.6	0.587	§
Energy intake, kcal/kg BW/day, mean (SD)	26.5 ± 5.8	28.0 ± 5.8	24.2 ± 5.8	<0.001	§
Protein intake, g/kg BW/day, median [IQR]	1.12 [0.97–1.28]	1.19 [1.05–1.36]	1.05 [0.92–1.17]	<0.001	*
Energy intake/Basal energy expenditure, median [IQR]	1.26 [1.11–1.38]	1.30 [1.18–1.41]	1.16 [1.00–1.30]	<0.001	*
Rehabilitation therapy, min/day, median [IQR]	137 [120–147]	137 [118–148]	137 [121–146]	0.402	*
FIM score on admission, median [IQR]					
Total FIM	73 [49–96.5]	70 [45–92]	78 [53–101]	0.088	*
Motor FIM	49 [28–68.5]	46 [26.25–65]	55 [35–71]	0.073	*
Cognitive FIM	24 [17–30.5]	24 [16–30]	24 [17–31]	0.342	*
Length of hospital stay, days, median [IQR]	124 [96–158]	130.5 [100–155.75]	120 [92.5–166.5]	0.626	*
FIM gain, median [IQR]	24 [15–37]	25.5 [15–38]	23 [15–36]	0.298	*
Motor FIM gain, median [IQR]	20 [11–30.5]	21 [12–31]	19 [10–27.5]	0.232	*

* Mann–Whitney U test; † Fisher’s exact test; ‡ Chi-square test; § *t* test; WMG: weight maintenance or gain group; WL: weight loss group; IQR: interquartile range; SD: standard deviation; CCI: Charlson comorbidity index; BMI: body mass index; EI/BEE: energy intake (kcal)/Basal energy expenditure (kcal); FIM: functional independence measure.

**Table 2 nutrients-14-00264-t002:** Multivariate analysis of motor FIM gain (model 1).

Factor	Standardized Coefficient	*p*-Value	VIF
Age	−0.125	0.037	1.526
Sex	0.031	0.590	1.449
Stroke type	0.009	0.878	1.305
Days from onset	−0.212	<0.001	1.449
BMI	0.133	0.011	1.142
Serum albumin level on admission, mg/dL	0.059	0.360	1.779
Serum creatinine level on admission, mg/dL	−0.027	0.638	1.417
Motor FIM at admission	−0.489	<0.001	2.317
CCI	−0.126	0.019	1.217
Dysphagia	−0.166	0.006	1.552
Length of stay	0.208	<0.001	1.404
Rehabilitation therapy, min/day	0.082	0.119	1.181
WMG	0.106	0.043	1.156

**Table 3 nutrients-14-00264-t003:** Multivariate analysis of motor FIM gain (model 2).

Factor	Standardized Coefficient	*p*-Value	VIF
Age	−0.197	0.003	1.869
Sex	0.018	0.756	1.442
Stroke type	0.008	0.885	1.305
Days from onset	−0.208	0.000	1.431
BMI	0.170	0.002	1.284
Serum albumin level on admission, mg/dL	0.037	0.559	1.747
Serum creatinine level on admission, mg/dL	−0.021	0.715	1.414
motor FIM at admission	−0.521	<0.001	2.374
CCI	−0.121	0.024	1.220
Dysphagia	−0.142	0.021	1.614
Length of stay	0.194	<0.001	1.420
Rehabilitation therapy, min/day	0.072	0.168	1.190
EI/BEE	0.169	0.005	1.529

## Data Availability

No new data were created or analyzed in this study. Data sharing is not applicable to this article.

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
