# Peer review of "Weight Change during the Early Phase of Convalescent Rehabilitation after Stroke as a Predictor of Functional Recovery: A Retrospective Cohort Study"

_nutrients, 2022, doi:10.3390/nu14020264_

Round 1

Reviewer 1 Report

The paper presents interesting factors regarding weight loss or gain during the subacute stage of post-stroke patients, although, some important features are missing from the research.
First of all, functional recovery shouldn't be assessed only by motor FIM.
It is not specified the time since stroke in the cohort, and also is not specified by timing what means "early stages of convalescence".
What exactly did the patients perform during PT and OT? In the statistical analysis (although good performed), the authors should replace units/day of rehabilitation to time ( minutes) and to detail what PT consisted of- because it is an essential feature regarding aerobic or anaerobic exercises, proprioception or coordination training, gait or stairs training.
The results section is missing the difference between admission and discharging regarding BMI and weight.
Also, the authors should include in their research aspects regarding other associated diseases, like diabetes, which implies a special diet.
From the research are also missing details regarding protein intake which can influence motor rehabilitation.
In line 158 please give details regarding the adjusted "other" factors.

Reviewer 2 Report

The paper is well written and important in clinical practice.

Just some minor concerns to improve introduction:

It would be important to analyze the role of some genetic factors, as the genetic variant of RANKL, as an independent risk factor for ischemic stroke [RANK/RANKL/OPG pathway: genetic association with history of ischemic stroke in Italian population European review for medical and pharmacological sciences 20, Issue 21, 1 November 2016, Pages 4574-4580] and of telomere [A review of telomere length in sarcopenia and frailty. Biogerontology, 2018, 19(3-4), pp. 209–221]. About Nutrients authors should cite the role of some of those involved in the process especially in elderly [Selenium Concentrations and Mortality Among Community-Dwelling Older Adults: Results from ilSIRENTE Study. Journal of Nutrition, Health and Aging, 2018, 22(5), pp. 608–612 - Myeloperoxidase levels and mortality in frail community-living elderly individuals. Journals of Gerontology - Series A Biological Sciences and Medical Sciences, 2010, 65 A(4), pp. 369–376].
